# Scorpion (*Hottentotta tamulus*) venom pre-exposure delays functional recovery in mice following peripheral nerve injury

Abbas Khan[1], Chand Raza[1]*, Muhammad Tariq Zahid[1], Hafiz Faseeh ur Rehman[2,3], Shahzad Sharif[4], Khajid Ullah Khan[1]

1 Department of Zoology, Faculty of Chemistry and Life Sciences, Government College University, Lahore, Pakistan, 2 Department of Anatomy and Histology, University of Veterinary and Animal Sciences, Lahore, Pakistan, 3 Lecturer Structure and Function, School of Veterinary Medicine, University of Central Lancashire, Preston, United Kingdom, 4 Department of Chemistry, Faculty of Chemistry and Life Sciences, Government College University, Lahore, Pakistan

* chandraza@gcu.edu.pk

## Abstract

Scorpion sting leads to profound challenges of central nervous system (CNS) impairments such as neuro-inflammation, unconsciousness, aberrant ion channels physiology, epilepsy and may become fatal due to heart failure. However persistence of *Hottentotta tamulus* venom in peripheral nerves and subsequent influence on regenerative process of injured peripheral nerve remains unknown. Current study reports the persistence of *H. tamulus* venom components 30-days following its intraperitoneal administration in sciatic nerves (SN) of mice pre-exposed through either a single-toxin exposure (STE) or multiple-toxin exposure (MTE). Of note, venom pre-exposure delays and compromises the sensori-motor functional recovery in STE and MTE mice following standard sciatic nerve crush injury. Histological investigations of regenerating SN and gastrocnemius muscles (GCM) 14-days post crush injury exhibited reduced myelination and limited numbers of motor axons in SNs and GCM of MTE mice, respectively. Consistently, a marked reduction in expression of regeneration-promoting markers including transcription factors (such as *Atf-3* and *c-Jun*), regeneration associated genes (such as *Sprr1a* and *Gap-43*) and ion channel proteins encoding genes (such as *Scn9a* and *Kcc2*) was observed in lumber dorsal root ganglia (DRG) and regenerating SN 14-days post crush injury. Collectively, this study reports the persistence and regeneration-inhibiting effects of *H. tamulus* venom in peripheral nerve of pre-exposed mice leading to compromised functional recovery.

## Introduction

Neuro-active components of scorpion venom affect the nervous system through a variety of ways, including central and peripheral neurotoxicity, targeting sodium (Na⁺)

**Data availability statement:** All relevant data are within the manuscript and its Supporting Information files.

**Funding:** The Office of Research, Innovation and Commercialization (ORIC), Government College University Lahore, Pakistan provided funds (Ref. # 152/ORIC/23) to PhD scholar Abbas Khan to support the research. The funders had no role in study design, data collection and analysis, decision to publish, or preparation of the manuscript.

**Competing interests:** Authors do not have any financial or other competing interests.

and potassium ($K^+$) ion channels, enhanced neurotransmitter release and abrupt behavioral patterns such as epileptic seizures [1,2]. Increasing reports suggested a huge increase in scorpion bite incidences and an estimated population of 2.3 billion is considered at risk of scorpion envenomation [3]. In Pakistan, scorpion sting incidence is poorly reported, however available data suggested thousands of victims experienced health challenges and sought medical attention for cure [4,5]. Venom is lipophilic in nature hence lipid-rich tissues like brain and nerves had a greater tendency to accumulate it [6,7]. Evidence suggested that peptide-based venom neurotoxins leave the circulatory system faster than other venom components and are reported to be present in organs like, kidney, heart, liver and lungs, pointing to their broader influences in affected individual [8]. Blood nerve barriers effectively restrict the infiltration of blood-borne molecules to peripheral nerves; however venom components such as ciguatoxins (marine neurotoxin in tainted fish) are reported to be present in the sciatic nerve after two months following envenomation. The extent of neurological impairments seemed chronic than gastrointestinal symptoms, and hence may remain unnoticed in PNS for prolonged periods, and are believed to have persistent neurological effects in persons [6].

Scorpion venom profoundly influence the nervous system homeostasis through inducing neuro-inflammation [9]. Neurotoxic peptides affect functioning of nervous system through altering the excitation and impulse conduction processes [10]. A disorganized release of neurotransmitters resulting in complex pattern of central and peripheral responses following envenomation is observed [11,12]. Neurotoxins are broadly characterized as alpha (comprising of short-chain peptides) and beta-toxins (comprising of long-chain peptides) interfered the functioning of voltage-gated $Na^+$ and $K^+$ channels, respectively [13]. Among 100 different components from *H. tamulus* venom, β-toxins (11 peptides) and α-toxins (45 peptides) are reported [14]. αKTx20.1 (peptide of *H. tamulus*) is reported to modulate vertebrate voltage-gated $K^+$ ion channels (Kv1.1 to Kv1.5) activity in *Xenopus laevis* oocyte [15]. Neurological dysfunctions are reported through venom induced increased $Na^+$ ions mobility, altering membrane permeability and leading to prolong neuronal firing [16]. Tb1, a β-neurotoxin from *Tityus bahiensis* venom, is reported for its action on $Na^+$ ion channels to reduce the current amplitude and shift to more hyperpolarized potentials [2]. Ts1, a homologue of Tb1, altered the cytokine level rather than damaging the neuron while Tb1 raised the hippocampal glutamate level and induced severe neuronal damage, suggesting functional differences among toxins. Furthermore, increased cytosolic $Na^+$ ion concentration in turn influenced the $Ca^{2+}$ signaling, altering the regulation of transcription factors, like nuclear factor kappa-light chain-enhancer of activated B cells (NF-κB), modulated the expression of neurotropic factor encoding gene like nerve growth factor (NGF) and activated T-cells [16,17]. Ion channel functioning and $Ca^{2+}$ signaling directly influenced regeneration of injured neurons [18,19], and hence venom components may have significant impact on nerve regeneration.

Severe peripheral nerve injuries (PNI) often lead to life-long disabilities, resulting in huge socio-economic burden [20,21]. Regeneration of injured peripheral nerve is mediated through changes in expression profiling of key regeneration-associated

genes (such as Jun proto-oncogene c-Jun (*c-Jun*), activating transcription factor-3 (*Atf-3*), small proline-repeat protein 1A (*Sprr1a*) and growth-associated protein-43 (*Gap-43*) [22]. Profound roles of ion channels in mediating neuronal outgrowth and guidance of regenerating axons following nerve injury are reported [19,23]. Of note, toxins such as ciguatoxins persisted for months in the nervous system and compromised the regenerative capacities of injured peripheral nerve to restore motor functions [6]. It suggests, envenomation may lead to extended risk duration for affected individuals to experience prolong suffering times following PNI.

Previous studies provide undeniable data of scorpion venom mediated alterations in neuronal functions through altering ion-channels permeability, inducing neuro-inflammation and dysregulated release of neurotransmitters [12,16,17,24], however the roles of scorpion envenomation on regenerative capacities of injured peripheral nerve and its influence on onset of functional recovery are lacking. In the current study, *H. tamulus* venom intoxication in mice for its persistence in peripheral nervous system (PNS), influence on regeneration of sciatic nerve following injury, modulation of key regeneration-associated genes, histological aspect of regenerating nerve and onset of injury-induced lost sensorimotor functions are investigated. This study provides novel insights into persistence of scorpion venom components in peripheral nerves and relevant chronic susceptibility for compromised regenerative capacities.

## Materials and methods

### Scorpion (*Hottentotta tamulus*) collection, rearing and venom extraction

The scorpions were collected during three months duration (June to August, 2022) from graveyards, bushes, wall crevices and low-hill mountains of Kotli District of Azad Jammu & Kashmir with the help of portable UV-light source (SOGO-JPN-139) and metallic tongs. The scorpions were transported in ventilated polypropylene based square containers containing gravel-sand based 2 cm thick bedding layer to the Neurobehavioral laboratory at Department of Zoology, GC University Lahore, Pakistan. The scorpions were identified morphologically and 20 adult scorpions (*Hottentotta tamulus*) of 8.0 ± 0.5 cm length were housed separately in standard ventilated polypropylene based semi-transparent terrarium (of 15 × 7 × 7 cm L × W × H dimensions) containing gravel-sand based bedding. Houseflies, cockroaches and grasshoppers were randomly provided to the scorpion terraria twice a week and scorpions had unlimited access to water during the study duration [14].

Venom extraction involved restraining the scorpion (food-deprived for 7 days) by applying an adhesive tape covering cephalothorax-abdomen area while placed on a polypropylene tray, however the tail was immersed in normal saline to facilitate the electrical conductance. The telson was positioned in an Eppendorf tube and a gentle electrical stimulation of 25V for 5 ± 1 seconds duration was applied so that scorpion droplets fell into the collecting Eppendorf tube, and stored at −20°C [14,25]. Venom extraction from each scorpion was done after every 15 days (for 3 times) and all the collected samples were pooled. The collected venom was then mixed with distilled water in 1:2 ratios (venom: d.$H_2O$) and lyophilized at −80°C (Freeze Dryer, LFD-BT-104 LABOCON). The lyophilized venom was then kept at −20°C for later use [14].

### Swiss albino mice grouping and dosing

Adult healthy male (8 weeks old) Swiss albino mice (25 ± 5 grams weight) were obtained from the animal housing facility at Government College University Lahore and kept at the mice rearing room located at the Department of Zoology Government College University Lahore, Pakistan under standard conditions (25 ± 2°C, 40–60% humidity, 12:12hrs light:dark cycle) to acclimatize for a week. All the animal related experiments were performed with prior approval from the Institutional Bioethics Committee on the use of laboratory animals (Ref. # GCU-IIB-2408).

To investigate persistence of *H. tamulus* venom components in the peripheral nerve, mice were randomly assigned to three groups (n = 6/group), namely (i) control group (CL) receiving 5% DMSO (10 ml/kg, intraperitoneally), as venom was dissolved in 5% DMSO as reported previously [26] (ii) single-toxin exposure group (STE) receiving 1 mg/kg venom intraperitoneally [26] and (iii) multi-toxin exposure group (MTE) receiving 1 mg/kg venom, once a week for 4 consecutive

weeks. Lyophilized scorpion venom dose of 1 mg/kg (in 5%DMSO containing distilled water) was selected for intraperitoneal administration based on previous study. The selected dose was safe for Swiss albino mice and caused no observable adverse effects on mice health during study duration [26].

To investigate the influence of *H. tamulus* venom components on the sensory-motor functional recovery following sciatic nerve crush injury, a separate batch of adult male Swiss albino mice was obtained and were assigned to three groups (n = 6/group) as described above (i.e., CL, STE and MTE groups). Following the venom injection in STE mice and last venom dose injection in MTE, the mice were subjected to left unilateral sciatic nerve crush injury.

The behavioral study lasted till 29 days following sciatic nerve crush injury, as previous studies widely report regain of full sensorimotor functions in healthy mice following 4-weeks of crush injury [6,22]. Mice in all study groups were humanely euthanized following last behavioral assessment, with ketamine (100 mg/kg, intraperitoneal) administration, by cervical dislocation after 29 days of sciatic nerve crush surgery. No animal died before the study endpoint.

## Mice surgery for sciatic nerve crush

Mice in all groups were subjected to unilateral left sciatic nerve crush injury, as described previously [6]. Briefly, mouse was anesthetized with intraperitoneal administration of cocktail of ketamine/xylazine (100 mg/kg/ 10 mg/kg). Mouse was placed on a surgery platform, under a dissecting microscope (model) facing dorsal side upwards. Fur on the left thigh region was removed with an electric shaver and cotton swab (soaked in 75% ethanol) was gently applied to sanitize the skin. A small incision (1 cm in length) on left thigh region was made with autoclaved scissors, the muscles covering the sciatic nerve (SN) were displaced gently to expose the nerve. A crush injury was performed with smooth forceps for 15 seconds at the level of external rotator muscle, distal to the sciatic notch. The nerve was covered with muscles and skin was sutured with 5−0 silk-derived sutures (B.T. Sutures). The mice were then shifted back to their respective cages and monitored until they started performing normal activities. The surgery was performed blinded to the mice grouping. Mice in all groups were monitored for their health on daily basis by observing the shiny fur (indicating grooming), mice movements and eyes.

## Detection of venom components from sciatic nerves

High performance liquid chromatography (HPLC) based venom components detection from sciatic nerves of mice was performed on 30th day following envenomation. Briefly, mice were anesthetized with ketamine (100 mg/kg, intraperitoneally) and euthanized by cervical dislocation. Immediately following cervical dislocation, the mice were decapitated and SN was harvested with autoclaved surgical tools. Harvested nerves were washed in ice-cold phosphate buffered saline (PBS), sciatic nerves were grounded in pestle motor using liquid nitrogen to obtain nerve homogenates. The homogenate was shifted into 1.5mL centrifuge tube suspended in 1 ml PBS (5%). The mixtures obtained from the nerves and crude venom were then filtered using syringe filter (HYDOCS, NY 0.22µm pore size) and the filtrate was then used for the HPLC analysis. HPLC analysis of crude scorpion venom mixed with double distilled water (1:2), centrifuged (13000 rpm) for 20 minutes at 4˚C and filtered through 0.22 pore-sized syringe filter [27] and sciatic nerve homogenates of CL, STE and MTE mice were performed using a reverse phase HPLC. The fractions were obtained using LC 20T Diode Array Detector system (Shimadzu, Japan) when samples were loaded into C18 column (250 mm × 4.6mm) with 100Å pore size and particle size 5µm. A linear gradient 0–67% solution B in solution A set at 200 µL/minute for 75 minutes (Solution A: 0.1% aqueous TFA, Solution B: 90% acetonitrile/water acidified with 0.09% TFA) [28] was set to obtain the fractions. The obtained chromatographs were compared for the venom components from crude venom and nerve homogenates from CL, STE and MTE mice groups.

## Behavioral assays

Different behavioral assays were carried out to check the functional recovery of sciatic nerve after SN crush injury. The mice were observed for the behavioral data from the 3rd day post injury (dpi) till the 30th dpi (Humane endpoints in Supporting Information file). These assays include the pinprick, toe spread, sciatic functional index and beam walk assays to

evaluate the functional recovery. All experiments involving mice handling and behavioral assessment were performed by trained and experienced researcher.

**Pin prick assays.** Mice were restrained on wire mesh for about 15–20 mints. After the habituation Austerlitz insect pin was used to record the data. Austerlitz insect pin was used to touch the lower planter surface of left hind paw of the mice. The lateral plantar surface of left hind paw was divided in 5 regions from heel to lateral most toe tip, ranging A to E. The insect pin was touched one by one to all the five regions of the hind paw mice. The brisk paw withdrawal reflex at site A upon pinprick stimulation was scored as 1, at B the score 2, and so on, and maximum score was 5 if mice showed paw-withdrawal reflex at toe-tip (site E). Failure of mice paw withdrawal at site A was scored as 0 as reported previously [29].

**Toe spread assay.** Toe spread assay was done to monitor the motor functional recovery. Mouse was wrapped in a piece of cloth in such a way that its tail was held in fingers of the observer and its caudal half including portion of belly and hind legs remained uncovered in cloth. A small tap was made on thumb of the holding hand of the observer and toe spread reflex of mouse was observed. The wide-spread toes with four clear spaces in-between and sustained reflex for 2 seconds was scored as 2. The partial recovery was shown by the intermediate spreading of the digits, in which mice demonstrated toe spread, however failed to sustain it (score 1). The club shaped phenotypes of hind paw and no toe spread reflex was scored as 0 [29]. The test was performed daily from 3rd day to 30th day after surgery.

**Sciatic functional index.** Motor functional recovery following sciatic nerve crush was quantitatively monitored through sciatic functional index (SFI) test [30–33]. The planter surface of both hind paws was painted with water soluble, non-toxic red China Ink. The T-maze (a straight corridor of 2.5 feet length, 8 cm width and 9 cm tall boundary walls with a 2 feet terminal corridor held at 90 degrees in T-shape with same width and boundary wall dimensions) used for this experiment was floored with a white paper strip of 8 cm width. The mice with painted hind-paws were then allowed to walk freely in the T-maze so that foot prints could be obtained. Three pairs of the foot-prints (of both ipsilateral and contralateral sides) were selected for SFI calculations. The measurements of foot-prints were made with Vernier calipers. Sciatic functional index values were calculated by using the following formula:

$$SFI = \left[ \left( \frac{NPL - EPL}{EPL} \right) + \left( \frac{ETS - NTS}{NTS} \right) + \left( \frac{EIS - NIS}{NIS} \right) \right] \times 73$$

Following lengths were calculated:
Paw length (PL: from heel to the tip of the middle finger)
Toe spread (TS: distance between the 1st and the 5th digit tips)
Inner Toe Spread (ITS: distance between the 2nd and 4th digit tips)
Whereas, "E" represents experimental (ipsilateral) while "N" represent normal (contralateral uninjured) foot-prints.

The SFI score around −100 depicts the total loss of motor function. Less negative SFI scores (approaching 0) indicate return of injury induced lot function.

## Histological analysis

Histological analysis of cardinal parameters (including myelination) of regenerating nerves of rodents is reported at 14 days following nerve injuries [34,35], therefore in current study the histological assessment was made 14 days post sciatic nerve crush injury. The gastrocnemius muscles and sciatic nerves were obtained after 14th day of SN crush injury from CL, STE and MTE groups for the histological examinations. The obtained tissues were fixed in 4% paraformaldehyde (PFA) solution for 24 hours at 4°C. Tissues were embedded in molten paraffin wax. 13 mm segments of SN from injury site to distal side were harvested and processed. 5 μm thick sections at 10 mm distal to injury site were obtained through microtome and mounted on slides. The SN sections were processed for methylene blue staining as described previously [36]. 20 μm thick sections from gastrocnemius muscle (GCM) belly were obtained and processed for silver nitrate staining, with some

modifications [37]. The prepared slides were then analyzed for the motor nerve re-innervation in GCM and myelination of SN with camera-fitted microscope.

The extent of regenerating SN myelination was measured using ImageJ (Version 1.54, NIH, USA). 30 non-overlapping images were selected randomly per group and myelin density was calculated in ImageJ with little modifications in described method by Jakic et al. [38]. Briefly, all the selected images were processed for optimal signal to noise ratio by applying uniform settings of "level adjustment" of 8-bit grayscale modified images in ImageJ. Area, mean and integrated density was recorded to measure the extent of myelination [39].

## Gene expression studies

Selected regeneration associated genes (RAGs) and ion channel protein encoding genes were investigated for their modulated expression in regenerating SNs and lumber (L4, L5 & L6) dorsal root ganglia (DRG) after 14th day of SN crush injury. Previous study reported the expression level monitoring following 14 days post sciatic nerve lesion yielded insightful expression data at transcript and protein levels of candidate regeneration promoting genes and proteins [40]. Briefly, the mice were terminally euthanized by cervical dislocation following deep anesthesia (100 mg/kg ketamine i.p.). The euthanized mice were decapitated, blood was drained and mice were positioned (dorsal side facing the observer) on a surgery pad under dissecting microscope. Skin of the dorsal side was excised, spinal cord was exposed by cracking and removing the spines and DRG in the lumber region were exposed, harvested in Eppendorf tubes and flash-frozen in liquid nitrogen. SNs were exposed and 15 mm distal segments from injury site were harvested and flash frozen [41,42]. Harvested tissues were homogenized in TriZol reagent (Invitrogen) by frequent micro-pipetting and phenol-chloroform based total RNA samples were isolated [43]. Briefly, chloroform was added to the TriZol containing homogenates, centrifuged at 12,000g for 15 minutes at 4°C. The aqueous layer containing RNA was transferred to a new Eppendorf tube followed by the addition of Isopropanol, centrifuged (12,000g for 10 minutes at 4°C) again and the pellet was washed with 75% ethanol. The RNA was suspended in deionized water and concentration was estimated with nanodrop.

Reverse transcription to generate cDNA ($OD_{260/280}$ ratio 1.8–2.0) from isolated RNA was achieved through RevertAid cDNA synthesis kit (Thermo Fisher Scientific, USA). Briefly, RNA (1 µg) was reverse transcribed to yield cDNA. The reaction mixture comprising of 1 µg RNA and random hexamer was maintained at 65°C for 5 minutes, with a brief incubation in ice (1 minute). Reaction mixture was prepared from reaction buffer, dNTPs, reverse transcriptase (RevertAid) and RNase to make final volume of 10 µl. The PCR tubes containing 10 µl of reaction mixture was sustained in the thermocycler (Quantarus) at 95°C for 5 minutes, 42°C for 60 minutes and 70°C for 5 minutes. The cDNA thus obtained was diluted with RNase/DNase free water (1:4) and stored at −20°C.

Quantitative real-time polymerase chain reaction (RT-PCR) was performed with SYBRgreen probe (Thermo Fisher Scientific, USA) by using 96-well plates (Low-profile, Bio-Rad, USA). Primers specific to *Gapdh* (Glyceraldehyde 3-phosphate dehydrogenase), *Scn9a* (sodium voltage-gated channel alpha subunit 9)*, Kcc2* (potassium-chloride cotransporter 2)*, c-Jun* (Jun proto-oncogene)*, Pgc-1α* (peroxisome proliferator-activated receptor gamma coactivator-1 alpha), *Stat3* (signal transducer and activator of transcription 3)*, Sox-11* (SRY-box transcription factor 11)*, Gap43* (growth associated protein 43), *Sprr1a* (small proline rich protein 1A) and *Atf3* (activating transcription factor 3) were used for quantification studies, designed using the Primer–BLAST from NCBI and Primer3 web [44] (Table 1). Afterwards, the Ct values were then analyzed using the $2^{-\Delta\Delta Ct}$ analysis with *Gapdh* as the reference control for normalization [45].

## Statistical analysis

The data obtained from the behavior tests was recorded and analyzed using GraphPad Prism software (version 5.0). Two-way ANOVA was applied to the behavioral data. Histological quantification for myelination and RT-qPCR data was analyzed by one-way ANOVA. Bonferroni post-hoc analysis. All the data were expressed as mean±SEM and values were considered significant when p<0.05.

 

**Table 1. Primer sequences and genes for RT-qPCR analysis.**

| Genes | | Sequence (5' to 3') |
|---|---|---|
| Gapdh (Glyceraldehyde 3-phosphate dehydrogenase) | Forward | CATGGCCTTCCGTGTTCCTA |
| | Reverse | CCTGCTTCACCACCTTCTTGAT |
| Atf3 (activating transcription factor 3) | Forward | CCAGGTCTCTGCCTCAGAAG |
| | Reverse | CATCTCCAGGGGTCTGTTGT |
| c-Jun (Jun proto-oncogene) | Forward | ACATGCTCAGGGAACAGGTG |
| | Reverse | TCAAAACGTTTGCAACTGCTG |
| Gap-43 (growth associated protein 43) | Forward | GTTTCCTCTCCTGTCCTGCT |
| | Reverse | CCACACGCACCAGATCAAAA |
| Sprr1a (small proline rich protein 1A) | Forward | GCTGTCTATCCTGCTTATGAGTC |
| | Reverse | CTTTGGGCAATGTTAAGAGGC |
| Scn9a (sodium voltage-gated channel alpha subunit 9) | Forward | CTACGCCATCTTTGGGATGT |
| | Reverse | TCAGGAGGTGCACTGTTGAG |
| Kcc2 (potassium-chloride cotransporter 2) | Forward | TTTGCTGCTCCTGTACGATG |
| | Reverse | GTGCCCAGGTAGAAGCAGAG |

## Results

### HPLC profiling

Persistence of scorpion venom derivatives was investigated following four weeks of its intraperitoneal administration in healthy mice. Mice sciatic nerve homogenates were analyzed through HPLC technique (S1 Fig in S1 File). The HPLC chromatograms showed presence of scorpion toxin components in single toxin-exposed group (STE) and multiple toxin-exposed groups (MTE) when compared with HPLC chromatogram of crude venom (Table 2).

### Venom intoxication compromises functional recovery

Sensory functional recovery was monitored through pinprick assay. Following SN crush injury, the CL group along with intoxicated groups (STE & MTE) showed complete loss of sensory function during 1st week. The CL and STE group showed an early onset of recovery on 7th and 8th day respectively after injury. Full recovery was gained on 24th dpi and 28th dpi in CL and STE groups, respectively. However, MTE group showed a significantly delayed onset of pinprick response initiated on 13th dpi, compared to control, thus hindering the sensory functional recovery. Importantly, MTE compromised the sensory functional recovery to greater extent in 33% mice population, as 2 out of 6 mice failed to achieve full sensory functional recovery (Fig 1A).

Motor functional recovery assessment was made by analyzing the re-innervation of motor axons in the tiny foot muscles controlling the movements of the toes. Toe spread reflex is an indication of the establishment of functional synapses controlling contraction of toe muscles. A delayed onset of motor functional recovery in MTE (13dpi) comparing to STE (10dpi) and CL (8dpi) was observed. Similarly, all mice in CL group demonstrated sustained toe spread 19dpi, 80% mice in STE and MTE groups fully recovered till the study completion (29dpi), respectively (Fig 1B).

Sciatic Functional Index (SFI) is a non-invasive, reliable and quantitative test to observe regain of motor functional recovery. SFI calculations of foot prints of healthy mice prior to SN crush surgery revealed a SFI score close to 0, indicating normal motor functions. SN crush injury led to loss of normal gait function, as SFI score close to −100 was observed 3dpi in all mice. In the first week post-surgery, the SFI score of CL group dropped to below −50 indicating substantial motor nerves impairment followed by STE group with an intermediate decline in between the CL and MTE groups. The MTE group showed a greater decline in the SFI score falling below −75. However, the MTE group showed meager improvement in the

Table 2. Comparison of venom fractions detected in crude Venom and Sciatic nerve homogenates of Cl, STE and MTE mice.

| CL | Venom | STE | MTE |
|---|---|---|---|
| 14.5 | 32.2 | 32.4 | 32.3 |
| 15.3 | 32.6 | 32.9 | 32.5 |
| 23.3 | 34.2 | 33.9 | 32.5 |
| 23.4 | 35.5 | 35.1 | 32.9 |
|  | 52.5 | 36.4 | 33.3 |
|  | 54.0 | 41.3 | 33.6 |
|  | 55.0 | 57.3 | 34.1 |
|  | 62.9 |  | 35.0 |
|  | 63.0 |  | 35.6 |
|  | 69.0 |  | 35.8 |
|  | 71.7 |  | 36.2 |
|  |  |  | 36.5 |
|  |  |  | 51.5 |
|  |  |  | 55.5 |
|  |  |  | 56.3 |
|  |  |  | 62.8 |
|  |  |  | 63.9 |
|  |  |  | 64.3 |
|  |  |  | 68.9 |
|  |  |  | 71.5 |

Retention time (Unit:Minutes).

2nd and 3rd weeks. A significant difference in SFI scores was noted between the control and MTE groups during the first and third weeks following SN injury, highlighting the delayed improvement in injury-induced lost gait functions (Fig 1C). However, all the mice in CL, STE and MTE groups showed comparable SFI scores following 4th week of SN crush injury.

## Venom intoxication reduces re-myelination and limits motor axonal regeneration

To our notion and consistent with delayed onset of functional recovery in MTE and STE mice groups must be reflected in the extent of regeneration of injured SN and motor axonal re-innervation in gastrocnemius (GCM) muscles. Thus, injured SNs and GCMs of mice were harvested after 14dpi and subjected to methylene blue staining to monitor extent of re-myelination and silver nitrate staining to observe regenerated motor axons, respectively. Methylene blue stained regenerating SN sections revealed a lesser degree of re-myelination in MTE group, comparing to either STE or CL groups (Fig 2A) (Fig 2). A quantitative estimation of intensity of methylene blue stained area in SN sections of CL, STE and MTE groups unveiled a significant (p < 0.01) decrease in MTE group in sharp contrast to either STE or CL groups (Fig 2B). The decreased extent of myelination in regenerating SN in MTE validates the delayed onset of sensory functional recovery.

Motor axonal re-innervation in GCM reflected the establishment of functional neuromuscular junctions to regain SN injury induced motor deficits. The GCM sections of CL mice showed presence of motor axons in multiple areas. A slight decline in motor axons appeared in STE group, however in MTE group a limited number of regenerating axons were observed (Fig 2C).

## Gene expression studies

Nerve injury incidences trigger the regenerative program of injured neurons for optimal functional recovery regains. Expression of regeneration associated genes (RAGs) such as *Small proline rich protein 1A* (*Sprr1A*) and *Growth*

 

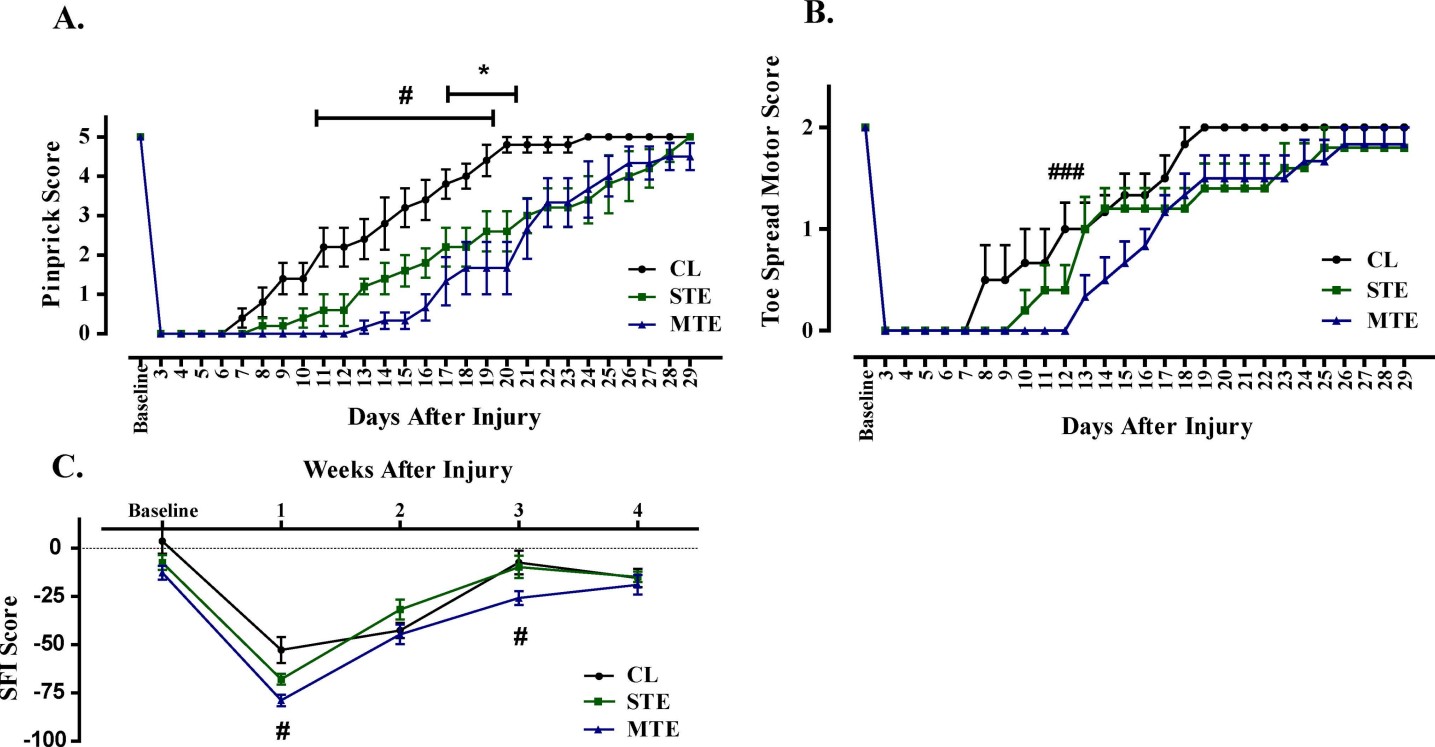

**Fig 1. Scorpion envenomation delayed the functional recovery onset in mice following sciatic nerve crush injury. A.** A delayed onset of sensory functional recovery in mice treated with single toxin exposure (STE) group (8dpi) and multi toxin exposure (MTE) mice (13dpi) comparing to control (CL) mice (7dpi), led to compromised full regain of sensory functions till the end of study duration (29dpi). **B & C.** Motor functional recovery in STE and MTE mice groups was hindered as depicted in lower toe spreading and more negative SFI scores and it was delayed comparing to CL mice. n = 5-6per group, data shown as mean±SEM. *p < 0.05 CL vs STE; #p < 0.05, ###p < 0.001 CL vs MTE. Two-way ANOVA with Bonferroni post-hoc analysis.

associated protein 43 (*Gap43*) is regulated by collaborative functions of transcription factors such as SOX-11, ATF-3 and c-JUN, and is upregulated following nerve injury [46–48]. Of note, ion channels are implicated in regenerative and physiological functions of neurons [49]. Therefore, dorsal root ganglia (DRG) and regenerating sciatic nerves from mice in CL, STE and MTE groups were harvested 14dpi and subjected to total RNA extraction, cDNA synthesis and reverse transcription-quantitative polymerase chain reaction (RT-qPCR) analysis for selected genes.

The expression levels of key transcription factors (*Atf-3* and *c-Jun*) were significantly (p < 0.001) reduced in lumber DRGs of STE and MTE mice comparing to those in CL mice (Fig 3), suggesting a decline in expression of associated RAGs. Similarly, regenerative marker genes *Sprr1a* and *Gap43* were significantly suppressed in MTE intoxicated mice, while in STE mice *Gap43* levels were significantly reduced but not the levels of *Sprr1a* expression. Sodium ion channel (Nav1.7) encoding gene *Scn9a* levels were suppressed significantly in STE but not in MTE, and expression levels of potassium/chloride (K⁺/Cl⁻) cotransporter encoding gene *Kcc2* were significantly downregulated in MTE mice, while *Kcc2* expression in STE mice remained undetected in lumber DRGs comparing to CL.

Similarly, in regenerating SN (14dpi) a sharp decline (p < 0.001) in *Atf-3* expression was observed in STE and MTE mice but the expression levels of *c-Jun* were reduced non-significantly in scorpion venom-injected groups. Consistent with the expression levels of *Gap-43* in lumber DRG, significant (p < 0.001) downregulated levels of *Gap-43* were observed in SNs of STE and MTE mice. However, the expression level fluctuations of *Sprr1a* in SNs of STE and MTE were not significant comparing to CL mice. The expression levels of *Scn9a* were reduced significantly (p < 0.05) in STE and MTE mice SNs comparing to SNs of CL mice. The expression of *Kcc2* in regenerating SNs of all groups remained undetected.

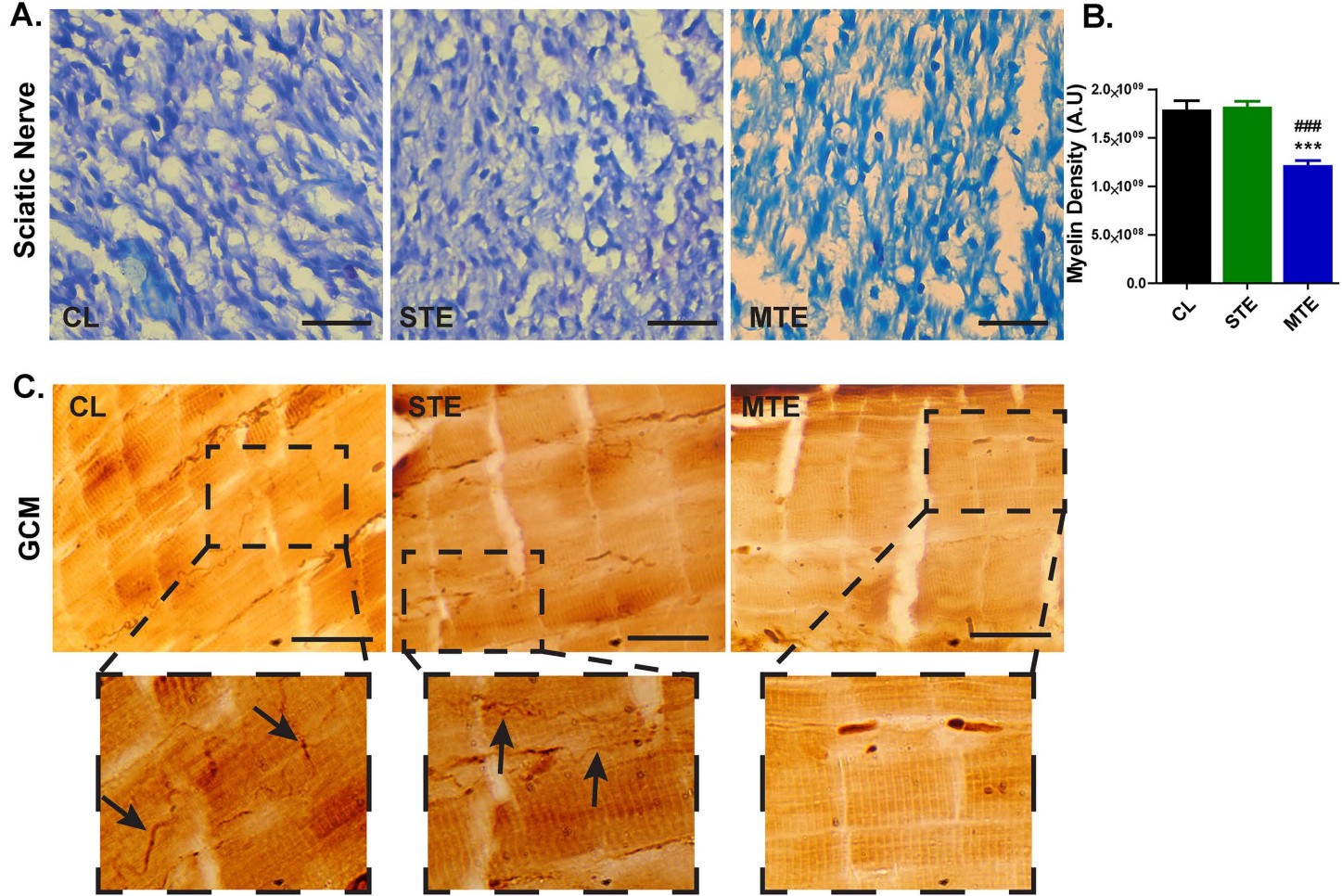

**Fig 2. Scorpion venom exposure limits regenerative capacities of injured sciatic nerve. A**. Histological observations of regenerating SN revealed a limited remyelination in MTE mice. Scalebar 5μm. **B.** Quantitative estimation of myelination revealed a significant reduction in MTE group comparing to either CL or STE mice. **C.** A higher number of regenerating motor axons in silver nitrate stained GCM sections were observed in CL and STE groups, however a limited number of motor axons were seen in MTE mice GCM sections. Scalebar 100μm. Data represented as mean ±SEM; n = 3mice/group; ***p < 0.001, CL vs MTE; ###p < 0.001 STE vs MTE. One-way ANOVA followed by Bonferroni post-hoc analysis.

Collectively, these results revealed that *H. tamulus* venom persisted in the sciatic nerves and have profound influence on limiting the regenerative capacities of injured neurons, through downregulating the regeneration-promoting genes and hence compromising the time to achieve functional recovery.

## Discussion

Current study demonstrated the persistence of *H. tamulus* venom components in the sciatic nerves of healthy mice after either a month of single safe-dose envenomation (in STE group) or a month after four safe-dose envenomation (in MTE group). Next, venom-induced delay in functional recovery of mice experiencing peripheral (sciatic) nerve crush injury was assessed through a battery of standard behavioral tests. Furthermore, histology of regenerating SN revealed compromised myelination, and limited numbers of regenerating motor axons in gastrocnemius muscle (14-days post crush injury) in MTE mice. Finally, expression profiling of regeneration-promoting transcription factors (such as *Atf-3* and *c-Jun*), and RAGs (such as *Sprr1a* and *Gap-43*) and ion channel encoding genes (such as *Scn9a* and *Kcc2*) implicated in

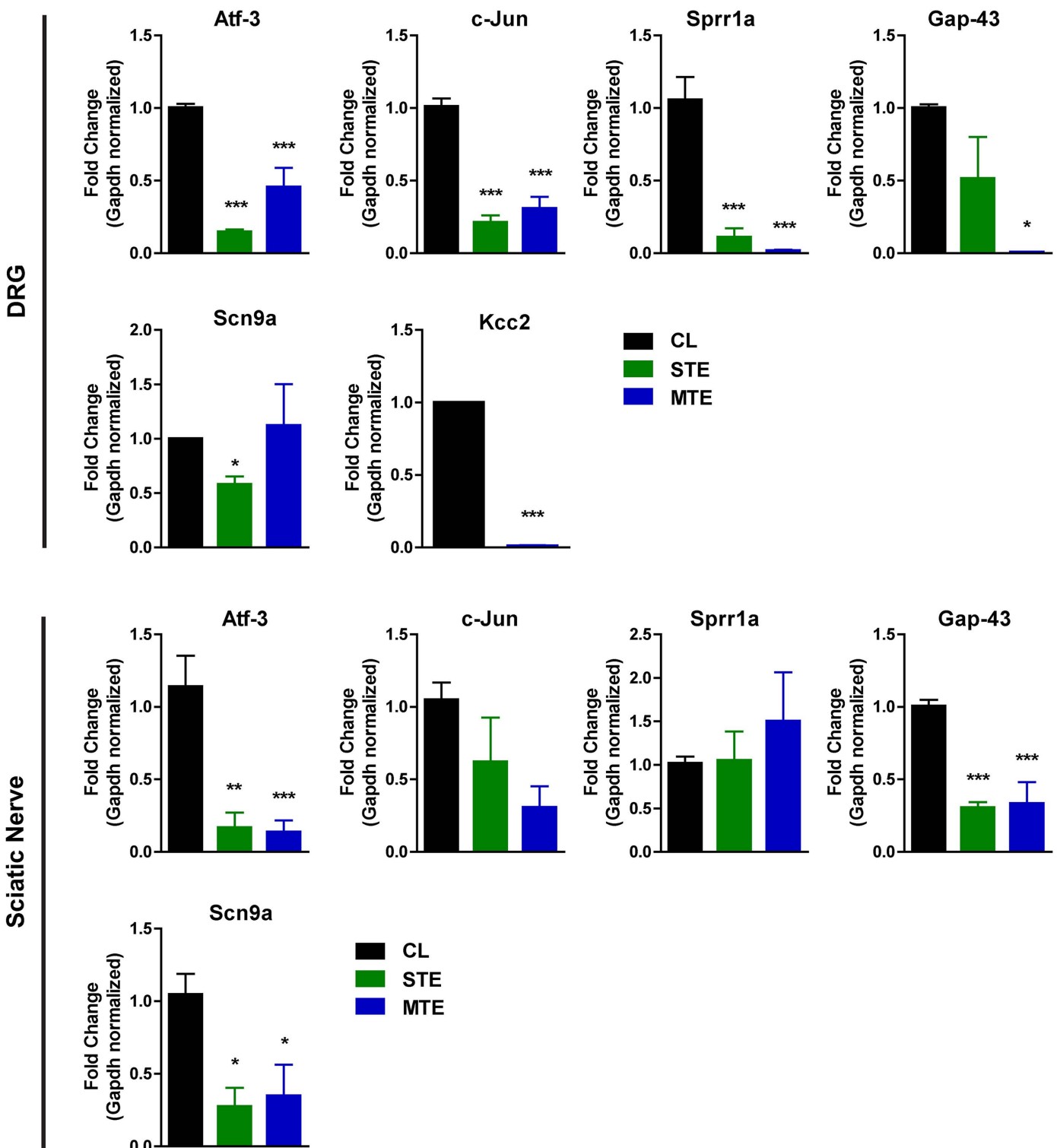

**Fig 3. Scorpion envenomation reduces the regenerative capacities of injured peripheral neurons and sciatic nerves through modulating expression of regeneration marker genes and ion channel encoding genes 14dpi.** Atf-3 and c-Jun are down-regulated in lumber DRGs. *Gap-43* and *Sprr1a* are downregulated in MTE mice while in STE only *Gap-43* is significantly downregulated, comparing to CL. Na ion channel encoding gene *Scn9a* is downregulated in STE, while K⁺/Cl⁻ cotransporter encoding gene *Kcc2* is downregulated in MTE mice. In regenerating SNs, a significant

downregulation of *Atf-3* was observed in STE and MTE, while expression levels of *c-Jun* fluctuated non-significantly. Similarly, the expression of *Gap-43* was downregulated in STE and MTE mice in SNs, while the levels of *Sprr1a* were fluctuated non-significantly in venom-injected mice groups. Data represented as mean±SEM, n=3 mice per group. *p<0.05; **p<0.01; ***p<0.001 CL vs venom-injected groups. Oneway ANOVA followed with Bonferroni posthoc analysis.

regeneration and axonal development were downregulated in STE and MTE dorsal root ganglia and SNs. These findings strengthen the notion of correlation of scorpion envenomation with compromised regenerative capacities of injured peripheral nerves. It may be valuable in future studies to investigate the key venom components interacting with the injured nerve tissue and to identify the therapeutic targets to design strategies aimed for accelerated axonal regeneration to achieve optimal functional recovery.

Scorpion (*H. tamulus*) envenomation is reported to have hazardous effects (including inflammation, burning pain, arrhythmia, seizures with unconsciousness leading to heart failure) within a couple of days to weeks of toxin exposure [50,51]. Several lines of evidence suggested the scorpion envenomation is linked with human neurological manifestations, such as damage to basal ganglia [52] and generation of destructive brain lesions causing epilepsy [53]. It is important to mention that the neurological disorder (epilepsy) persisted for months after a single scorpion toxin exposure, suggesting the prolong persistence of venom components in the tissue, probably in CNS [53]. Based on existing literature, we hypothesized that *H. tamulus* envenomation might induce neurological deficits in peripheral nerves and might influence the regenerative process following sciatic crush injury in mice. Therefore, in first line of our investigations we reasoned whether scorpion venom components could persist for sufficient time required to regain functional recovery (29-days) following peripheral (sciatic) nerve crush injury. Previously, a study revealed that *Hemiscorpius lepturus* venom labeled with $^{67}$Gallium radioisotope persisted in mice brain for a shorter duration (72 hours) following its administration [54]. However, so far no studies reported the persistence of *H. tamulus* venom components in CNS or PNS. Current study reports the persistence of *H. tamulus* venom components in SNs of healthy mice following one month after single- or multiple-toxin exposures. Importantly, HPLC analysis from SN homogenate of MTE mice demonstrated many eluting fractions coinciding with those of crude toxin, while a few fractions were coinciding between STE mice SN homogenates and crude toxin. This data agrees with our notion of possible modulation of regeneration process of injured peripheral nerve with the involvement of persistent toxin components.

In real world, scorpion envenomation at daily basis is not reported however, frequent scorpion sting incidences are reported over time. In current study, the aspect of repeated scorpion envenomation (in MTE group) was considered equally important along with single envenomation (STE mice). Weekly administration of venom safe dose (1 mg/kg) was hypothesized to mimic the real-life envenomation risks. Of note, a bi-weekly administration of toxin (affecting CNS neurons) is reported to cause high mortality [55], and excessive exposure of sub-lethal doses (0.01 mg/kg and 0.02 mg/kg) caused painful effects with frequent numbness signs in toxin injected paw [56]. Prolong exposure of toxin (exceeding 4-weeks duration) might lead to excessive functional adaptation and is less likely to occur in habitats. Importantly, venom components influence the activity of critical ion channels (such as Na$^+$ and K$^+$ ion channels) and frequent envenomation over longer duration of time could lead to neurotoxicity and cardiovascular collapse leading to animal mortality. Therefore, in current study a modest dose administration schedule once a week for 4 consecutive weeks was selected to maintain a consistent toxic burden, to estimate cumulative influence on nerve regeneration.

Restoration of sensory and motor functions following SN crush injury in adult mice completes in 29-days [6]. Pinprick stimulation to lateral plantar surface of ipsilateral paw following SN crush injury failed to evoke brisk paw withdrawal. However, during 2nd week of SN crush injury, mice started to respond pinprick stimulation as an indication of regeneration of sensory axons to appropriate dermatomes and thus functional recovery is expected upon successful regeneration of sensory axons till the distal most dermatome of lateral plantar paw skin [22]. A notable delay in onset of pinprick response in MTE mice was in agreement with our hypothesis of toxin-influenced modulation of regenerative capacities

of injured sensory neurons. Delay in onset of sensory functional recovery or failure to regain full sensory functions is associated with compromised regenerative capacities of DRG neurons [6] and dysfunctions of ion channels [57]. Crush injury induced loss of motor functions are expected to restore upon successful regeneration-promoting mechanisms operating in motor neurons that lead to axonal growth in correct guidance path, clearance of axonal debris of distal stump, successful reinnervation to muscles and finally re-establishment of functional neuromuscular junctions (NMJs) [22]. Restoration of gait functions and gripping functions of mice ipsilateral hind paw are result of motor axonal growth, optimal myelination and re-establishment of NMJs in the tiny muscles of lateral plantar region. Improper myelination, limited axonal growth capacities and failure to establishment of NMJs could result in motor functional deficits [58]. A delayed onset of motor functions (toe spreading motor response and SFI scores), limited myelination of regenerating nerve and reduced motor axon density in gastrocnemius muscles in current study signify the influence of persisted *H. tamulus* toxin fractions on restoration of injury induced lost motor functions and it is notable that small populations of STE and MTE mice groups failed to regain full motor functions indicating prolong suffering durations and chronic effects.

Injury to peripheral (sciatic) nerve triggers the transcriptional programs of associated sensory (soma of which reside in DRGs) neurons and local translation events of mRNAs in the regenerating axons as an innate regenerative response [59]. Therefore, in second line of our investigation we extracted lumber (L4, L5 & L6) DRGs and regenerating SN segments 14-dpi to monitor the expression levels of critically important transcription factors (*Atf-3* and *c-Jun*), RAGs (*Sprr1a* and *Gap-43*) and ion channel encoding genes (*Scn9a* and *Kcc2*) through RT-qPCR. It aimed to know the influence of persisted toxin components on innate regenerative capacities of injured neurons. A larger population of DRG neurons (large and small diameter soma, but not in intermediate size) expressed ATF-3 following sciatic nerve injury, similarly crush or transection SN injury the expression levels of GAP-43 and SPRR1A were observed in ipsilateral DRG neurons, but not in contralateral neurons [60,61]. Small proline rich repeat protein 1a is expressed in axotomized neurons and promotes axonal outgrowth). Overexpression of ATF-3 in mice increased the neurite outgrowth and elongation in DRG neurons and upregulated the expression of SPRR1A [62], similarly reduced *Atf-3* expression compromised the regenerative capacities of injured neurons [63]. Additionally, GAP-43 following its expression in DRGs is reported to transport rapidly to growth axonal distal segments, and evidence suggested its local translation in distal axonal segments [64], and its downregulation is associated with severe reduction in growth capacities of regenerating axons [65]. Compelling evidence reported c-Jun expression persisted in neurons during the entire course of regeneration [66] and is a master regulator of Schwann cell transition from myelinating to proliferating stage, modulates the production of trophic factors, extracellular matrix components, removal of debris from distal nerve stump and myelination around regenerating axons [67]. c-JUN dysregulation following nerve injury hinders the regeneration process and prolongs the recovery duration [68]. Current expression study, in agreement with behavioral and histological data reports a significant downregulation in expression of key transcription factors and RAGs in DRG and SNs of STE and MTE mice groups. Importantly, *c-Jun* and *Sprr1a* were significantly downregulated in lumber DRGs but not in regenerating SNs of venom-intoxicated mice. It might be owing to Schwann cell based expression of *c-Jun* in SNs [67] and regulation of *Sprr1a* by regulatory proteins in SNs other than *Atf-3* and *c-Jun* transcription factors [48,69].

Perturbed homeostasis of sodium, potassium and chloride ions in neurons is associated with reduced neurite outgrowth capacities of neurons *in vitro* and *in vivo*. A pacific fish poison, ciguatoxin, is known for its excessive sodium ion influx in cultured DRG neurons leading to limited neurite outgrowth [7]. Of note, a sub-optimal influx of ions in regenerating neurons modulates the axonal growth following peripheral nerve injury [70]. Similarly, blockage of voltage gated sodium channels in regenerating axons lead to reduced ATP production and degenerative processes [71]. A wider variety of *H. tamulus* venom components is reported to perturb ionic homeostasis of neurons through interfering the activities of ion channels [14]. In current study, a reduced expression of voltage gated sodium ion channel encoding gene *Scn9a* was observed in SNs of toxin-injected mice, however its expression was not significantly different in lumber DRGs of MTE mice. The

differences in gene expression outcomes may be due to tissue-specific responses to injury, varying regenerative capacities, and the venom's neurotoxic effects on ion channel regulation. Reduced expression of *Scn9a* was observed in lumber DRGs of STE mice, and expression of potassium/chloride co-transporter encoding gene *Kcc2* was downregulated in MTE mice, however its mRNA level was not detected in STE mice lumber DRGs. It suggested that the persisted toxin components in DRGs and SNs might lead to abrupt ionic homeostasis, resulting in profound influence on regenerative capacities following peripheral nerve injury.

Although current study unveils the detrimental effects of *H. tamulus* envenomation on regeneration capacities of injured peripheral (sciatic) nerve at behavioral, histological and mechanistic levels, however it do presents some limitations to validate the observed effects through further investigations. First, further investigation to identify the persistent toxin components through gas chromatography-mass spectrometry (GC-MS) or MALDI-TOFF based protein fractions characterization could provide further insights on key relevant components of venom mediating delay in functional recovery onset. Second, immune-histochemistry of gastrocnemius muscle for functional NMJs (α-bungarotoxin and neurofilament-200 co-staining), toluidine-blue staining of semi-thin sections of regenerating SNs for myelination may strengthen the notion. Third, western blot analysis for expression studies of selected transcription factor (ATF-3 and c-JUN), RAGs (SPRR1A and GAP-43) and ion-channel encoding proteins could strengthen the currently unveiled data.

## Conclusion

Taken together, it is clear that the *H. tamulus* toxin fractions in mice persisted in peripheral nerves, delayed the restoration of SN crush injury induced lost sensory and motor functions, reduced the myelination of regenerating SN and motor axonal regeneration in gastrocnemius muscle, and negatively modulated the expression of regeneration-promoting transcription factors, genes and key ion channel encoding genes. Current findings present the *H. tamulus* envenomation as a significant risk, affecting the sensori-motor functional regains following peripheral nerve injury.

## Supporting information

**S1 File. All supporting figures (Fig S1 and Fig S2) and Humane endpoints referenced in the article are included in this file.**
(DOCX)

## Author contributions

**Conceptualization:** Chand Raza, Muhammad Tariq Zahid.

**Formal analysis:** Chand Raza.

**Investigation:** Abbas Khan, Chand Raza, Muhammad Tariq Zahid, Hafiz Faseeh ur Rehman, Shahzad Sharif, Khajid Ullah Khan.

**Methodology:** Abbas Khan, Chand Raza, Muhammad Tariq Zahid, Hafiz Faseeh ur Rehman, Shahzad Sharif, Khajid Ullah Khan.

**Resources:** Chand Raza.

**Supervision:** Chand Raza, Muhammad Tariq Zahid.

**Validation:** Muhammad Tariq Zahid.

**Visualization:** Chand Raza.

**Writing – original draft:** Abbas Khan, Chand Raza, Muhammad Tariq Zahid, Hafiz Faseeh ur Rehman, Shahzad Sharif, Khajid Ullah Khan.

**Writing – review & editing:** Chand Raza.

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
