## [Editor Report · Decision Letter 0]

30 Jan 2025

Dear Dr. Chand Raza,

Thank you for submitting your manuscript to PLOS ONE. After careful consideration, we feel that it has merit but does not fully meet PLOS ONE’s publication criteria as it currently stands. Therefore, we invite you to submit a revised version of the manuscript that addresses the points raised during the review process.

We look forward to receiving your revised manuscript.

Kind regards,

Yoshihiro Sowa

Academic Editor

PLOS ONE

Journal Requirements:

3. We note that your Data Availability Statement is currently as follows: [All relevant data are within the manuscript and its Supporting Information files]

Additional Editor Comments (if provided):

Methods Section

The authors administered 1 mg/kg of the toxin intraperitoneally to Swiss albino mice. It is necessary to not only cite references but also explicitly explain the rationale behind this dosage.

Additionally, the reason for administering MTE once per week for four consecutive weeks should be clearly stated. Why was the treatment repeated exactly four times?

In the Mice Surgery for Sciatic Nerve Crush procedure, what was the rationale for covering the crushed site with muscle tissue after the injury? Please provide a clear explanation.

Results Section

The Methylene blue staining images of regenerating sciatic nerve (SN) sections appear to be of poor quality. This makes quantitative analysis difficult. Why were the images not presented in color?

A detailed explanation of how the quantification was specifically performed using ImageJ is required.

The GCM staining in Figure 2C also has quality issues. The staining sites are unclear, making interpretation difficult. Fluorescent immunostaining using a marker for axonal expression would be preferable. Alternatively, higher magnification images with enhanced contrast should be provided.

Furthermore, it would be beneficial to present additional stained images, even as Supplementary Data. The current data are insufficient to substantiate the authors' hypothesis.

---

## [Author Response · Author response to Decision Letter 1]

18 Mar 2025

All the comments raised by the editor are addressed in the revised manuscript. The details of editor's comments are provided in the "Response to Editor" file

---

## [Decision Letter · Decision Letter 1]

17 Jun 2025

Dear Dr. Raza,

Thank you for submitting your manuscript to PLOS ONE. After careful consideration, we feel that it has merit but does not fully meet PLOS ONE’s publication criteria as it currently stands. Therefore, we invite you to submit a revised version of the manuscript that addresses the points raised during the review process.

We look forward to receiving your revised manuscript.

Kind regards,

Olfa Chakroun-Walha

Academic Editor

PLOS ONE

Journal Requirements:

Reviewers' comments:

Reviewer's Responses to Questions

**Comments to the Author**

Reviewer #1: (No Response)

Reviewer #2: (No Response)

2. Is the manuscript technically sound, and do the data support the conclusions?

Reviewer #1: Yes

Reviewer #2: Yes

3. Has the statistical analysis been performed appropriately and rigorously?

Reviewer #1: Yes

Reviewer #2: Yes

4. Have the authors made all data underlying the findings in their manuscript fully available?

Reviewer #1: No

Reviewer #2: Yes

5. Is the manuscript presented in an intelligible fashion and written in standard English?

Reviewer #1: Yes

Reviewer #2: Yes

Reviewer #1: Khan and colleagues studied the effects of the scorpion Hottentotta tamulus venom in peripheral nerves in mice. They showed that the venom persists following injection and that compromises the recovery of nerves. The study is comprehensive and the results support the hypothesis. Specific comments are listed below.

Major comments:

1- Is there any previous evidence of scorpion venoms (from any specie) persistence in the nervous systems (as authors describe for ciguatoxins)?

2- Why was the intraperitoneal route chosen for this studies since most envenomations in human occur via subcutaneous or intramuscular routes?

3- Was the expression of Gapdh similar in all groups? This is important since this gene was used for normalization.

Minor comments:

Page 3, line 53: please correct “present” to “be present”.

Page 5: define PNS.

Page 8: correct “1.5μL” to 1.5mL

Page 14: correct “oneway” to “one-way”

Reviewer #2: Based on behavioral testing, it was discovered that scorpion venom delayed neural recovery, and the author also explored the expression levels of related effect genes. Below are the questions the author needs to address:

1. The HPLC results confirmed the presence of scorpion venom residues, but did not provide an accurate material analysis of the venom. Additionally, what are the characteristic components of scorpion venom used for detecting residues? This needs to be clarified.

2. The author conducted tests on the expression levels of related effect genes. Are there any other image-based supporting evidences (such as fluorescence, histochemistry, etc.) available to corroborate these findings?

3. Some spelling errors, such as "N represent" at line 225 of the R1 revision.

**Do you want your identity to be public for this peer review?** For information about this choice, including consent withdrawal, please see our Privacy Policy

Reviewer #1: **Yes: ** Ana T. A. Sachetto

Reviewer #2: No

---

## [Author Response · Author response to Decision Letter 2]

14 Jul 2025

Response to Reviewers

We are thankful to the reviewers for insightful comments on the research article entitled “Scorpion (Hottentotta tamulus) venom pre-exposure delays functional recovery in mice following peripheral nerve injury” submitted for publication to the PlosOne Journal.

The article is revised in light of the suggestions and comments of the reviewers, details of which are given below,

Reviewer 1:

General Comment: Khan and colleagues studied the effects of the scorpion Hottentotta tamulus venom in peripheral nerves in mice. They showed that the venom persists following injection and that compromises the recovery of nerves. The study is comprehensive and the results support the hypothesis. Specific comments are listed below.

General Response: We are encouraged to revise our manuscript by the very valid comments of the reviewer 1. Point-to-point comments of the reviewer are addressed below and appropriate changes are made in revised manuscript.

Comment: Is there any previous evidence of scorpion venoms (from any specie) persistence in the nervous systems (as authors describe for ciguatoxins)?

Response: currently, to our knowledge, there is no direct evidence of persistence of scorpion venom components in the nervous system for extended periods in the same manner as ciguatoxins, which are known to integrate into neural membranes and exert long-term effects (Au et al., 2016). However, several studies support the idea of functional persistence of scorpion venom effects on the nervous system for hours to few days, primarily due to slow clearance of certain neurotoxic peptides and their prolonged impact on ion channels and neuroimmune responses.

Tityus serrulatus venom was observed for neuro-excitatory and inflammatory responses in rodent models with some physiological and behavioral changes for up to 72 hours post exposure. However, no molecular retention was demonstrated in these studies, while suggesting a sustained downstream effect of venom exposure on neuronal function (Pessini et al., 2003, Nencioni et al., 2000, Van Fraga et al., 2015).

Potassium and sodium channels were highly expressed, aligning with scorpion neurotoxicity profiles. This support the idea of neurotoxic activity post-envenomation, however, structural persistence of scorpion venom components is unproven in contrast to ciguatoxin. Additionally, metalloproteinase and other bioactive peptides may contribute to tissue inflammation and prolonged biological effects reinforcing the observations of functional persistence in the nerve regeneration and gene expression studies (Costal-Oliveira et al., 2015).

Comment: Why was the intraperitoneal route chosen for this studies since most envenomations in human occur via subcutaneous or intramuscular routes?

Response: The intraperitoneal (IP) route was selected for the current study due to its effectiveness in ensuring rapid and systemic distribution of venom in the small animal model selected in the current study (Swiss albino mice). IP route is utilized because it allows the controlled delivery, reduced local tissue variability and reproducibility in dose-response outcomes in venom toxicology research. The IP route was chosen to enable consistent exposure across experimental groups and to better mimic the systemic effects of venom in controlled conditions.

Intraperitoneal injection of T. serrulatus and T. bahiensis scorpions induced electrographic and behavioral alterations, in previous studies, suggesting intraperitoneal route as a potential route of venom administration in pre-clinical studies (page no. 236, 3rd paragraph from (Nencioni et al., 2009)).

Comment: Was the expression of Gapdh similar in all groups? This is important since this gene was used for normalization.

Response: Yes, Gapdh expression was consistent across all experimental groups, as validated by quantitative polymerase chain reaction (qPCR) analysis by observing the Ct values. Gapdh showed least variation across control (CL), single toxin exposure (STE) and multiple toxin exposure (MTE) groups hence served as an internal control for normalizing expression levels of selected genes in the current study.

Comment: Page 3, line 53: please correct “present” to “be present”.

Response: Appropriate change is made in line with the reviewer’s comment.

Comment: Page 5: define PNS.

Response: Appropriate change is made in revised manuscript.

Comment: Page 8: correct “1.5μL” to 1.5mL

Response: Correction is made in revised manuscript.

Comment: Page 14: correct “oneway” to “one-way”

Response: Correction is made in the revised manuscript.

Reviewer 2:

General Comment: Based on behavioral testing, it was discovered that scorpion venom delayed neural recovery, and the author also explored the expression levels of related effect genes. Below are the questions the author needs to address:

General Response: We appreciate the comments of the reviewer 2 to improve the manuscript submitted for publication in the PlosOne journal. The manuscript is revised and the comments of the reviewer 2 are addressed below,

Comment: The HPLC results confirmed the presence of scorpion venom residues, but did not provide an accurate material analysis of the venom. Additionally, what are the characteristic components of scorpion venom used for detecting residues? This needs to be clarified.

Response: We acknowledge the limitation regarding the material characterization. The current study was designed to detect the residual components of scorpion venom based on established protocols. However, full molecular characterization was not conducted in the current work, the retained peaks match previously described low molecular weight toxins (6-8kDa) are possibly associated with neurotoxic effects. We are thankful for the reviewer to highlight this important point, however the characterization of scorpion venom components was not the objective of the research, as one of PhD scholar’s PhD research project involves the LC-MS based characterization of the scorpion venom and hence it was decided not to go for venom component characterization in current study. Further, we have mentioned this limitation in last paragraph of the discussion sections (line 504). However, we feel the suggested comment needs to be addressed through a separate study following ethical and departmental approvals. And the identified component of scorpion venom persisted in sciatic nerves might be explored for its regeneration-modulating effects through a separate in vivo study.

Comment: The author conducted tests on the expression levels of related effect genes. Are there any other image-based supporting evidences (such as fluorescence, histochemistry, etc.) available to corroborate these findings?

Response: We are thankful to the reviewer for highlighting an important point of validating qPCR data on protein level through immune-histochemistry. In current study, in addition to the gene expression data, histological analysis of the sciatic nerve tissue was performed using methylene blue staining in an attempt to observe the myelination around regenerating axons to correlate the compromised expression of regeneration-associated genes with the sub-optimal myelination in nerves of intoxicated mice. However, lack of fluorescence microscope, antibodies and relevant logistics restrained us to perform the mentioned fluorescence based detection of protein products of the regeneration-associated genes (Gap43, Atf3, Sox11 etc). Therefore, we regret to have such relevant data in current article.

In the manuscript, last paragraph of discussions section reports the limitation of the current study and in line with the reviewer’s comment, the limitation of unavailability of Western blot equipment and facilities are mentioned (Line 510 of the manuscript) which restrained us to explore the expression levels of regeneration-associated genes at protein levels.

Comment: Some spelling errors, such as "N represent" at line 225 of the R1 revision.

Response: “N” represent Normal paw (Contralateral- uninjured) foot-prints. Appropriate revision is made in revised version.

AU, N. P., KUMAR, G., ASTHANA, P., TIN, C., MAK, Y. L., CHAN, L. L., LAM, P. K. & MA, C. H. 2016. Ciguatoxin reduces regenerative capacity of axotomized peripheral neurons and delays functional recovery in pre-exposed mice after peripheral nerve injury. Sci Rep, 6, 26809.

COSTAL-OLIVEIRA, F., GUERRA-DUARTE, C., CASTRO, K. L., TINTAYA, B., BONILLA, C., SILVA, W., YARLEQUE, A., FUJIWARA, R., MELO, M. M. & CHAVEZ-OLORTEGUI, C. 2015. Serological, biochemical and enzymatic alterations in rodents after experimental envenomation with Hadruroides lunatus scorpion venom. Toxicon, 103, 129-34.

NENCIONI, A. L. A., CARVALHO, F. F., LEBRUN, I., DORCE, V. A. C. A. & SANDOVAL, M. R. L. 2000. Neurotoxic effects of three fractions isolated from Tityus serrulatus scorpion venom. Pharmacology & toxicology, 86, 149-155.

NENCIONI, A. L. A., LOURENCO, G. A., LEBRUN, I., FLORIO, J. C. A. & DORCE, V. A. 2009. Central effects of Tityus serrulatus and Tityus bahiensis scorpion venoms after intraperitoneal injection in rats. Neuroscience letters, 463, 234-238.

PESSINI, A. C., DE SOUZA, A. M., FACCIOLI, L. H., GREGóRIO, Z. M. & ARANTES, E. C. 2003. Time course of acute-phase response induced by Tityus serrulatus venom and TsTX-I in mice. International immunopharmacology, 3, 765-774.

VAN FRAGA, I. T., LIMBORCO-FILHO, M., LIMA, O. C. O., LACERDA-QUEIROZ, N., GUIDINE, P. A. M., MORAES, M. F. D., ARAúJO, R. N., MORAES-SANTOS, T., MASSENSINI, A. R. & ARANTES, R. M. E. 2015. Effects of tityustoxin on cerebral inflammatory response in young rats. Neuroscience Letters, 588, 24-28.

---

## [Decision Letter · Decision Letter 2]

5 Aug 2025

Scorpion (Hottentotta tamulus) venom pre-exposure delays functional recovery in mice following peripheral nerve injury

PONE-D-25-01116R2

Dear Dr. Raza,

We’re pleased to inform you that your manuscript has been judged scientifically suitable for publication and will be formally accepted for publication once it meets all outstanding technical requirements.

Kind regards,

Olfa Chakroun-Walha

Academic Editor

PLOS ONE

Additional Editor Comments (optional):

Reviewers' comments:

Reviewer's Responses to Questions

**Comments to the Author**

Reviewer #1: All comments have been addressed

2. Is the manuscript technically sound, and do the data support the conclusions?

Reviewer #1: Yes

3. Has the statistical analysis been performed appropriately and rigorously?

Reviewer #1: Yes

4. Have the authors made all data underlying the findings in their manuscript fully available?

Reviewer #1: Yes

5. Is the manuscript presented in an intelligible fashion and written in standard English?

Reviewer #1: Yes

Reviewer #1: (No Response)

**Do you want your identity to be public for this peer review?** For information about this choice, including consent withdrawal, please see our Privacy Policy

Reviewer #1: **Yes: ** Ana T. A. Sachetto

---

## [Editor Report · Acceptance letter]

PONE-D-25-01116R2

PLOS ONE

Dear Dr. Raza,

I'm pleased to inform you that your manuscript has been deemed suitable for publication in PLOS ONE. Congratulations! Your manuscript is now being handed over to our production team.

Kind regards,

on behalf of

Pr Olfa Chakroun-Walha

Academic Editor

PLOS ONE